# Joint Sub-bands Learning with Clique Structures for Wavelet Domain Super-Resolution

**Zhisheng Zhong[1]    Tiancheng Shen[1,2]    Yibo Yang[1,2]    Chao Zhang[1,*]    Zhouchen Lin[1,3]**

[1]Key Laboratory of Machine Perception (MOE), School of EECS, Peking University
[2]Academy for Advanced Interdisciplinary Studies, Peking University
[3]Cooperative Medianet Innovation Center, Shanghai Jiao Tong University
{zszhong, tianchengshen, ibo, c.zhang, zlin}@pku.edu.cn

## Abstract

Convolutional neural networks (CNNs) have recently achieved great success in single-image super-resolution (SISR). However, these methods tend to produce over-smoothed outputs and miss some textural details. To solve these problems, we propose the Super-Resolution CliqueNet (SRCliqueNet) to reconstruct the high resolution (HR) image with better textural details in the wavelet domain. The proposed SRCliqueNet firstly extracts a set of feature maps from the low resolution (LR) image by the clique blocks group. Then we send the set of feature maps to the clique up-sampling module to reconstruct the HR image. The clique up-sampling module consists of four sub-nets which predict the high resolution wavelet coefficients of four sub-bands. Since we consider the edge feature properties of four sub-bands, the four sub-nets are connected to the others so that they can learn the coefficients of four sub-bands jointly. Finally we apply inverse discrete wavelet transform (IDWT) to the output of four sub-nets at the end of the clique up-sampling module to increase the resolution and reconstruct the HR image. Extensive quantitative and qualitative experiments on benchmark datasets show that our method achieves superior performance over the state-of-the-art methods.

## 1   Introduction

Single image super-resolution (SISR) is to reconstruct a high-resolution (HR) image from a single low-resolution (LR) image, which is an ill-posed inverse problem. SISR has gained increasing research interest for decades. Recently, convolutional neural networks (CNNs) [6, 25, 32] significantly improve the peak signal-to noise ratio (PSNR) in SISR. These networks commonly use an extraction module to extract a series of feature maps from the LR image, cascaded with an up-sampling module to increase resolution and reconstruct the HR image.

The quality of extracting features will seriously affect the performance of the HR image reconstruction. The main part of extraction module used in modern SR networks can be primarily divided into three types: conventional convolution layers [23], residual blocks [9] and dense blocks [10].

Conventional convolution has been widely considered by scholars since AlexNet [20] won the first prize of ILSVRC in 2012. The first model using conventional convolution to solve the SR problem is SRCNN [6]. After that, many improved networks such as FSRCNN [7], SCN [36], ESPCN [28] and DRCN [18] also use conventional convolution and achieve great results. Residual block [9] is an improved version of the convolutional layer, which exhibits excellent performance in computer vision problems. Since it can enhance the feature propagation in networks and alleviate the vanishing-

---

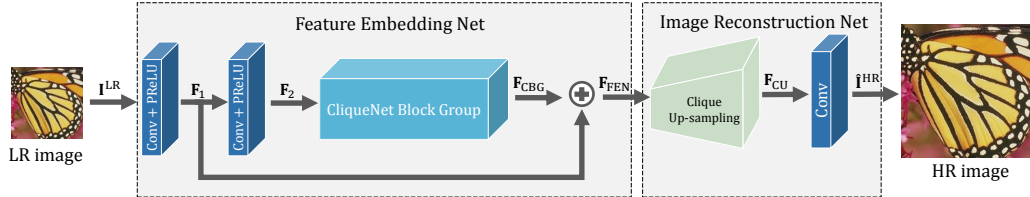

Figure 1: The architecture of the proposed Super-Resolution CliqueNet (SRCliqueNet).

gradient problem, many SR networks such as VDSR [17], LapSRN[22], EDSR [25] and SRResNet [24] import residual blocks and exhibit improved performances.

To make use of the skip connections used in residual blocks, Huang et al. proposed the dense block [10] further. A dense block builds more connections among layers to enlarge the information flow. Tong et al. [35] proposed SRDenseNet using dense blocks, which boosts the performance further more.

Recently, Yang et al proposed a novel block called the clique block [38], where the layers in a block are constructed as a clique and are updated alternately in a loop manner. Any layer is both the input and the output of another one in the same block so that the information flow is maximized. The propagation of a clique block contains two stages. The first stage does the same thing as a dense block. The second stage distills the feature maps by using the skip connections between any layers, including connections between subsequent layers.

A suitable up-sampling module can further improve image reconstruction performance. The up-sampling modules used in modern SR networks to increase the resolution can also be primarily divided into three types: interpolation up-sampling, deconvolution up-sampling and sub-pixel convolution up-sampling.

Interpolation up-sampling was first used in SRCNN [6]. At that time, there were no effective implementations of module that can make the output size larger than the input size. So SRCNN used pre-defined bicubic interpolation on input images to get the desired size first. Following SRCNN using pre-interpolation, VDSR [17], IRCNN [42], DRRN [31] and MemNet [32] used different extraction modules. However, this pre-processing step increases computation complexity because the size of feature maps is multiple.

Deconvolution proposed in [39, 40] can be seen as multiplication of each input pixel by a filter, which could increase the input size if the stride $s > 1$. Many modern SR networks such as FSRCNN [7], LapCNN [22], DBPN [8] and IDN [14] got better results by using deconvolution as the up-sampling module. However, the computation complexity of forward and back propagation of deconvolution is still a major concern.

Sub-pixel convolution proposed in [28] aims at accelerating the up-sampling operation. Unlike previous up-sampling methods that change the height and width of the input feature maps, sub-pixel convolution implements up-sampling by increasing the number of channels. After that sub-pixel convolution uses a periodic shuffling operation to reshape the output feature map to the desired height and width. ESPCN [28], EDSR [25] and SRMD [42] used sub-pixel convolution to achieve good performances on benchmark datasets.

These above-mentioned networks tend to produce blurry and overly-smoothed HR images, lacking some texture details. Wavelet transform (WT) has been shown to be an efficient and highly intuitive tool to represent and store images in a multi-resolution way [26, 30]. WT can describe the contextual and textural information of an image at different scales. WT for super-resolution has been applied successfully to the multi-frame SR problem [4, 16, 27].

Motivated by the remarkable properties of clique block and WT, we propose a novel network for SR called SRCliqueNet to address the above-mentioned challenges. We design the res-clique block as the main part of the extraction module to improve the network's performance. We also design a novel up-sampling module called clique up-sampling. It consists of four sub-nets which use to predict the high resolution wavelet coefficients of four sub-bands. Since we consider the edge feature properties of four sub-bands, four sub-nets can learn the coefficients of four sub-bands jointly. For magnification factors greater than 2, we design a progressive SRCliqueNet upon image pyramids [1]. Our proposed network achieves superior performance over the state-of-the-art methods on benchmark datasets.

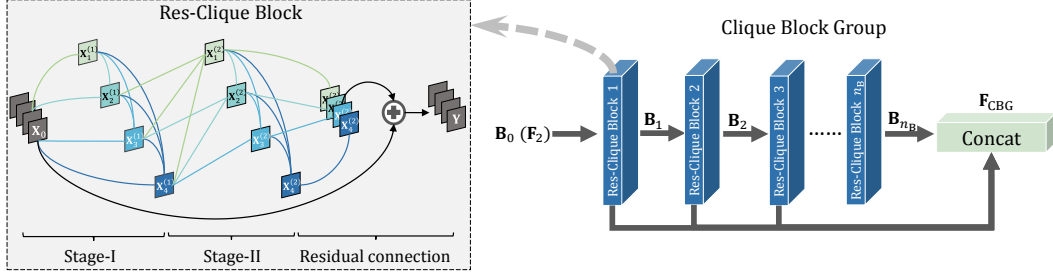

Figure 2: The illustrations of the res-clique block (left) and the clique block group (right).

# 2 Super-Resolution CliqueNet

In this section, we first overview the proposed SRCliqueNet architecture, then we introduce the feature embedding net (FEN) and the image reconstruction net (IRN), which are the key parts of SRCliqueNet.

## 2.1 Network architecture

As shown in Figure 1, our SRCliqueNet mainly consists of two sub-networks: FEN and IRN. FEN represents the LR input image as a set of feature maps. Note that FEN does not change the size $(h, w)$ of the input image, where $h$ and $w$ are the height and the width, respectively. IRN up-samples the feature map got by FEN and reconstructs the HR image. Here we denote $\mathbf{I}^{\mathrm{LR}} \in \mathbb{R}^{3 \times h \times w}$ as the input LR image and $\mathbf{I}^{\mathrm{HR}} \in \mathbb{R}^{3 \times rh \times rw}$ as the ground truth HR image, where $r$ is the magnification factor.

## 2.2 Feature Embedding Net

As shown in the left part of Figure 1, FEN starts with two convolutional layers. The first convolutional layer tries to increase the number of channels of input, which can be added with the output of the clique block group via the skip connection. The clique block group will be introduced immediately. The skip connection after the first convolutional layer has been widely used in SR networks [14, 24, 25]. The output of the first convolutional layer is $\mathbf{F}_1 \in \mathbb{R}^{nlg \times h \times w}$, where $n$ is the number of clique blocks that follow, $l$ is the number of layers in each clique block and $g$ is the growth rate of each clique block. The second convlutional layer tries to change the number of channels so that they can fit the input of clique block group. The output of the second convolutional layer is $\mathbf{F}_2 \in \mathbb{R}^{lg \times h \times w}$.

The illustrations of res-clique block and clique block group are shown in Figure 2. We choose clique block as our main feature extractor for the following reasons. First, a clique block's forward propagation contains two stages. The propagation of first stage does the same things as dense block, while the second stage distills the feature further. Second, a clique block contains more skip connections compared with a dense block, so the information among layers can be more easily propagated. We add a residual connection to the clique block, since the input feature contains plenty of useful information in terms of SR problem. We call such kind of clique block as the res-clique block.

Suppose a res-clique block has $l$ layers and the input and the output of the res-clique block are denoted by $\mathbf{X}_0 \in \mathbb{R}^{lg \times h \times w}$ and $\mathbf{Y} \in \mathbb{R}^{lg \times h \times w}$, respectively. The weight between layer $i$ and layer $j$ is represented by $\mathbf{W}_{ij}$. The feed-forward pass of the clique block can be mathematically described as the following equations. For stage one, $\mathbf{X}_i^{(1)} = \sigma(\sum_{k=1}^{i-1} \mathbf{W}_{ki} * \mathbf{X}_k^{(1)} + \mathbf{W}_{0i} * \mathbf{X}_0)$, where $*$ is the convolution operation, $\sigma$ is the activation function. For stage two, $\mathbf{X}_i^{(2)} = \sigma(\sum_{k=1}^{i-1} \mathbf{W}_{ki} * \mathbf{X}_k^{(2)} + \sum_{k=i+1}^{l} \mathbf{W}_{ki} * \mathbf{X}_k^{(1)})$. For residual connection, $\mathbf{Y} = [\mathbf{X}_1^{(2)}, \mathbf{X}_2^{(2)}, \mathbf{X}_3^{(2)}, ..., \mathbf{X}_l^{(2)}] + \mathbf{X}_0$, where $[\cdot]$ represents the concatenation operation.

Then we combine $n_{\mathrm{B}}$ res-clique blocks into a clique block group. The output of a clique block group makes use of features from all preceding res-clique blocks and can be represented as $\mathbf{B}_i = \mathcal{H}_{\mathrm{RCB}_i}(\mathbf{B}_{i-1}), i = 1, 2, 3, ..., n_{\mathrm{B}}, \mathbf{B}_i \in \mathbb{R}^{lg \times h \times w}$, where $\mathbf{B}_i$ is the output and $\mathcal{H}_{\mathrm{RCB}_i}$ is the underlying mapping of the $i$-th res-clique block. Since $\mathbf{F}_2$ is the input of the first res-clique block, we have $\mathbf{B}_0 = \mathbf{F}_2$. $\mathbf{F}_{\mathrm{CBG}} = [\mathbf{B}_1, \mathbf{B}_2, ..., \mathbf{B}_n] \in \mathbb{R}^{nlg \times h \times w}$ is the output of clique block group. Finally, the output of FEN is a summation of $\mathbf{F}_{\mathrm{CBG}}$ and $\mathbf{F}_1$, that is $\mathbf{F}_{\mathrm{FEN}} = \mathbf{F}_{\mathrm{CBG}} + \mathbf{F}_1$.

## 2.3 Image Reconstruction Net

Now we present details about IRN. As shown in the right part of Figure 1, IRN consists of two parts: a clique up-sampling module and a convolutional layer which is used to reduce the number of feature maps to reconstruct the HR image with 3 channels (RGB).

The clique up-sampling module showing in Figure 3 is the most significant part of IRN. It is motivated by discrete wavelet transformation (DWT) and clique block. It contains four sub-nets, representing four sub-bands denoted by LL, HL, LH and HH in the wavelet domain, respectively. Previous CNNs for wavelet domain SR [11, 21] ignore the relationship among the four sub-bands. The LL block represents low-pass filtering of the original image at half the resolution. The output feature maps of FEN encode the essential information in the original LR image. So we use the output feature $\mathbf{F}_{\text{FEN}}$ to learn the LL block firstly. We represent the number of channels of input feature maps by $c$, then $\mathbf{F}_{\text{FEN}} \in \mathbb{R}^{c \times h \times w}$, $c = nlg$. This process can be written as

$$\mathbf{F}_{\text{LL}}^{(1)} = \mathcal{H}_{\text{LL}}^{(1)}(\mathbf{F}_{\text{FEN}}), \tag{1}$$

where $\mathcal{H}_{\text{LL}}^{(1)}$ denotes the learnable non-linear function of the LL block for the first step. The HL block shows horizontal edges, mostly. In contrast, the LH block mainly contains vertical edges. As illustrated in the left part of Figure 4, we take an image from Set5 [3] as an example. Both the HL and LH blocks can be learned from the LL block and the feature $\mathbf{F}_{\text{FEN}}$, written as

$$\mathbf{F}_{\text{HL}}^{(1)} = \mathcal{H}_{\text{HL}}^{(1)}([\mathbf{F}_{\text{FEN}}, \mathbf{F}_{\text{LL}}^{(1)}]), \quad \mathbf{F}_{\text{LH}}^{(1)} = \mathcal{H}_{\text{LH}}^{(1)}([\mathbf{F}_{\text{FEN}}, \mathbf{F}_{\text{LL}}^{(1)}]), \tag{2}$$

where $\mathcal{H}_{\text{HL}}^{(1)}$ and $\mathcal{H}_{\text{LH}}^{(1)}$ denote the learnable function to construct the HL and the LH blocks for the first step. The HH block finds edges of the original image in the diagonal direction. Also shown in the left part of Figure 4, the HH block looks similar to the LH and the HL blocks, so we suggest that using LL, HL, LH blocks and the output feature map of FEN could learn the HH block easier than using the feature map alone. We formulate it as

$$\mathbf{F}_{\text{HH}}^{(1)} = \mathcal{H}_{\text{HH}}^{(1)}([\mathbf{F}_{\text{FEN}}, \mathbf{F}_{\text{LL}}^{(1)}, \mathbf{F}_{\text{HL}}^{(1)}, \mathbf{F}_{\text{LH}}^{(1)}]). \tag{3}$$

We name the above-mentioned operations as the sub-band extraction stage. We also plot four histograms at the right part of Figure 4 to prove that the sub-band extraction stage is effective. We apply DWT to 800 images from DIV2K [25] which we use as our training dataset in our experiments and plot histograms of four sub-bands' DWT coefficients of these images. From Figure 4, we find that the distributions of LH, HL and HH blocks are similar to each other. So it is reasonable to use the HL and LH blocks to learn the HH blocks.

The four sub-bands are followed by a few residual blocks after the sub-band extraction stage. Due to that high frequency coefficients may be more difficult to learn than low frequency coefficients, we use different numbers of residual blocks for different sub-bands. We denote the numbers of residual blocks of each sub-band as $n_{\text{LL}}, n_{\text{HL}}, n_{\text{LH}}$ and $n_{\text{HH}}$, respectively. we update each sub-band by the following equation

$$\mathbf{F}_{\text{LL}}^{(2)} = \mathcal{H}_{\text{LL}}^{(2)}(\mathbf{F}_{\text{LL}}^{(1)}), \ \mathbf{F}_{\text{HL}}^{(2)} = \mathcal{H}_{\text{HL}}^{(2)}(\mathbf{F}_{\text{HL}}^{(1)}), \ \mathbf{F}_{\text{LH}}^{(2)} = \mathcal{H}_{\text{LH}}^{(2)}(\mathbf{F}_{\text{LH}}^{(1)}), \ \mathbf{F}_{\text{HH}}^{(2)} = \mathcal{H}_{\text{HH}}^{(2)}(\mathbf{F}_{\text{HH}}^{(1)}), \tag{4}$$

where $\mathcal{H}_{\text{LL}}^{(2)}, \mathcal{H}_{\text{HL}}^{(2)}, \mathcal{H}_{\text{LH}}^{(2)}$ and $\mathcal{H}_{\text{HH}}^{(2)}$ represent the residual learnable function of for four sub-bands, respectively. We name the above-mentioned operations as the self residual learning stage.

After the operations of the self residual learning stage, IRN enters the sub-band refinement stage. At this stage, we use the high frequency blocks to refine the low frequency blocks, which is an inverse process of the sub-band extraction stage. Concretely, we use the HH block to learn the LH and the HL blocks, represented as

$$\mathbf{F}_{\text{LH}}^{(3)} = \mathcal{H}_{\text{LH}}^{(3)}([\mathbf{F}_{\text{HH}}^{(3)}, \mathbf{F}_{\text{LH}}^{(2)}]), \quad \mathbf{F}_{\text{HL}}^{(3)} = \mathcal{H}_{\text{HL}}^{(3)}([\mathbf{F}_{\text{HH}}^{(3)}, \mathbf{F}_{\text{HL}}^{(2)}]), \tag{5}$$

where $\mathcal{H}_{\text{LH}}^{(3)}$ and $\mathcal{H}_{\text{HL}}^{(3)}$ represent the learnable function of sub-band refinement stage for the LH and HL blocks, respectively. For the unification of representations, we define $\mathbf{F}_{\text{HH}}^{(3)} = \mathbf{F}_{\text{HH}}^{(2)}$. In a similar way, we update $\mathbf{F}_{\text{LL}}$ by the following equation

$$\mathbf{F}_{\text{LL}}^{(3)} = \mathcal{H}_{\text{LL}}^{(3)}([\mathbf{F}_{\text{HH}}^{(3)}, \mathbf{F}_{\text{LH}}^{(3)}, \mathbf{F}_{\text{HL}}^{(3)}, \mathbf{F}_{\text{LL}}^{(2)}]). \tag{6}$$

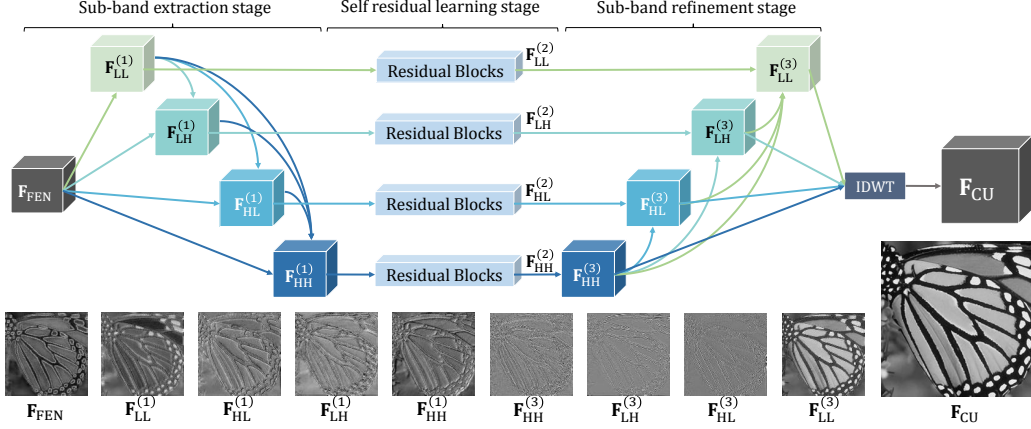

Figure 3: The architecture of clique up-sampling module and the visualization of its feature maps.

Then we apply IDWT to these four blocks, we choose the simplest wavelet, Haar wavelet, for it can be computed by deconvolution operation easily. The dimensions of all blocks are the same. They are all $p \times h \times w$, where $p$ represents the number of feature maps produced by each sub-net. So the output of clique up-sampling module is $\mathbf{F}_{\text{CU}} = \text{IDWT}([\mathbf{F}_{\text{LL}}^{(3)}, \mathbf{F}_{\text{HL}}^{(3)}, \mathbf{F}_{\text{LH}}^{(3)}, \mathbf{F}_{\text{HH}}^{(3)}]) \in \mathbb{R}^{p \times 2h \times 2w}$. At last, the output of clique up-sampling module is sent to a convolutional layer, which is used to reduce the number of channels and get the predicted HR image $\hat{\mathbf{I}}^{\text{HR}}$. We call the up-sampling module as clique up-sampling for the following reasons. First, the connection patterns of these two modules are consistent. Both of clique block and clique up-sampling use dense connections among sub-bands/layers. Second, the forward propagation mechanisms of these two modules seem to be similar, that is, both the two modules update the output of sub-bands/layers stage by stage. Since both the extraction module and the up-sampling module relate to clique, we call our network as Super Resolution CliqueNet (SRCliqueNet in short).

## 2.4 Comparison between clique block and clique up-sampling

Although we call the block and the up-sampling module as clique block and clique up-sampling, respectively, there are many differences between these two modules. Concretely, the number of sub-bands/layers of clique up-sampling is fixed to four because of the formula of IDWT. In contrast, the layer number of clique block is not constrained. Clique up-sampling has three stages to update the output of each sub-band/layer. The clique block, by contrast, does not have a stage that can update the output by its own layer alone. Since we consider the edge feature properties of all sub-bands, the HL block mostly shows horizontal edges. In contrast, the LH block mainly contains vertical edges. The outputs of these two blocks seem to be "orthogonal". So there may be no connection between the second and the third sub-bands/layers in clique up-sampling module. At last, the outputs of these two modules are quite different. To be more specific, the output of a clique block is the concatenation form of the output of all layers, which makes it have more channels. The output of clique up-sampling is the output of all layers after IDWT, which increases the resolution.

## 2.5 Architecture for magnification factor $2^{\mathbf{J}}\times$

Till now, we have introduced the network architecture for magnification factor $2\times$. In this subsection, we propose SRCliqueNet's architecture for magnification factor $2^J\times$, where $J$ is the total level of the network. Image pyramid [1] has been widely used in computer vision applications. LAPGAN [5] and LapSRN [22] used Laplacian pyramid for SR. Motivated by these works, we import image pyramid to our proposed network to deal with magnification factors at $2^J\times$. As shown in the left part of Figure 5, our model generates multiple intermediate SR predictions in one feed-forward pass through progressive reconstruction. Due to our cascaded and progressive architecture, our final loss consists of $J$ parts: $L = \sum_{j=1}^{J} L_j$. We use the bicubic down-sampling to resize the ground truth HR image $\mathbf{I}^{\text{HR}}$ to $\mathbf{I}^j$ at level $j$. Following [14, 25], we use mean absolute error (MAE) to measure the performance of reconstruction for each level: $L_j = \text{mean}(|\mathbf{I}^j - \hat{\mathbf{I}}^j|)$, where $\hat{\mathbf{I}}^j$ is the predicted HR image at level $j$.

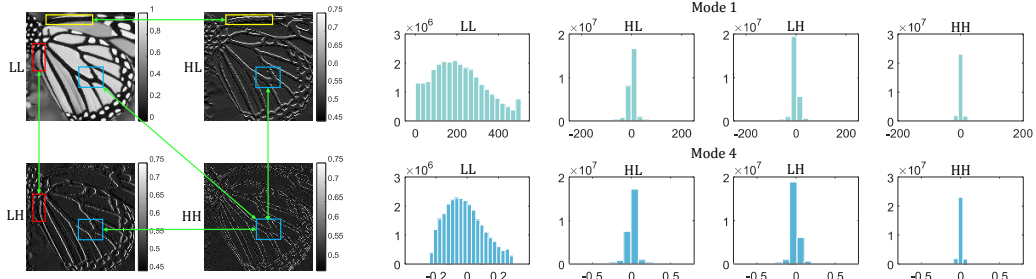

Figure 4: Left: The illustration of four sub-bands edge features relationships. Right: The histograms of four sub-bands' coefficients over 800 images from DIV2K [33]. Top right: Do DWT on original images. Bottom right: Do DWT on images preprocessed with mode 4 described in Section 3.1.

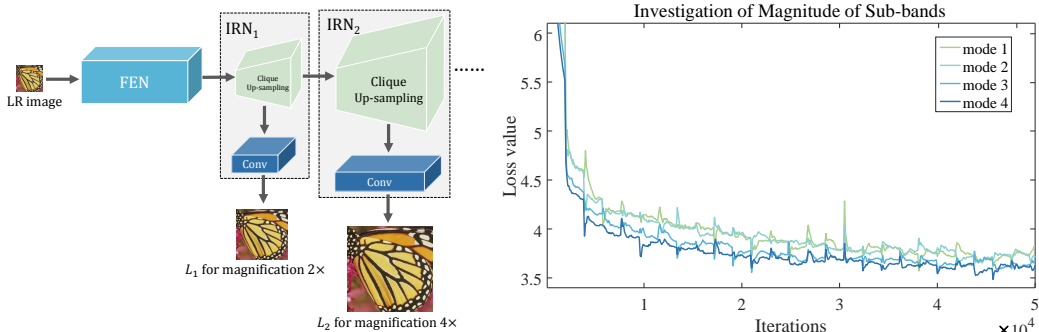

Figure 5: Left: The SRCliqueNet architecture with magnification a factor $4\times$. Right: the performances of input images transformed with four modes.

# 3 Experiments

## 3.1 Implementation and training details

**Model Details.** In our proposed SRCliqueNet, we set $3 \times 3$ as the size of most convolutional layers. We also pad zeros to each side of the input to keep size fixed. We also use a few $1 \times 1$ convolutional layers for feature pooling and dimension reduction. The details of our SRCliqueNet's setting are presented in Table 1. In Table 1, $n_B$ represents the number of clique blocks. $l$ and $g$ represent the number of layer and the growth rate in each clique block, respectively. The numbers of input and output channels of clique up-sampling module are denoted as $c$ and $p$, respectively. $n_{LL}, n_{LH}, n_{HL}$ and $n_{HH}$ represent the number of residual blocks in the four sub-bands. Unlike most CNNs for computer vision problems, we avoid dropout [29], batch normalization[15] and instance normalization [13], which are not suitable for the SR problem, because they reduce the flexibility of features [25].

**Datasets and training details.** We trained all networks using images from DIV2K [33] and Flickr [25]. For testing, we used four standard banchmark datasets: Set5 [3], Set14 [41], BSDS100 [2] and Urban100 [12]. Following settings of [25], we used a batch size of 16 with size $32 \times 32$ for LR images, while the size of HR images changes according to the magnification factor. We randomly augmented the patches by flipping horizontally or vertically and rotating $90°$. We chose parametric rectified linear units (PReLUs) as the activation function for our networks. The base learning rate was initialized to $10^{-5}$ for all layers and decreased by a factor of 2 for every 200 epochs. The total training epoch was set to 500. We used Adam [19] as our optimizer and conducted all experiments using PyTorch.

**Magnitude of sub-bands.** As mentioned above, our clique up-sampling module has four sub-nets and every sub-net has connection with the other sub-nets. Since the feature maps of one sub-band are learned from some other sub-bands', the magnitude of each sub-band block should be similar to others in order to get full use of each sub-net. As shown in Figure 4, the histograms of DWT coefficients of original images are at the top right part. The coefficients' magnitude of the LL sub-band is quite different from the other three's, which may make training process difficult. So we want to transform

Table 1: Details of our proposed SRCliqueNet for magnification factors $2\times$ and $4\times$. CBG represents Clique Block Group and CU represents Clique Up-sampling.

| Models | CBG | | | CU$_1$ | | | CU$_2$ | | |
|---|---|---|---|---|---|---|---|---|---|
| SRCliqueNet($2\times$) | $n_{\text{B}}$ | $l$ | $g$ | $c$ | $n_{\text{LL}}$ | $n_{\text{HL}}$ | | | |
| | | | | 1920 | 2 | 3 | | ✗ | |
| | 15 | 4 | 32 | $p$ | $n_{\text{LH}}$ | $n_{\text{HH}}$ | | | |
| | | | | 480 | 3 | 4 | | | |
| SRCliqueNet($4\times$) | $n_{\text{B}}$ | $l$ | $g$ | $c$ | $n_{\text{LL}}$ | $n_{\text{HL}}$ | $c$ | $n_{\text{LL}}$ | $n_{\text{HL}}$ |
| | | | | 2400 | 2 | 3 | 600 | 2 | 3 |
| | 15 | 4 | 32 | $p$ | $n_{\text{LH}}$ | $n_{\text{HH}}$ | $p$ | $n_{\text{LH}}$ | $n_{\text{HH}}$ |
| | | | | 600 | 3 | 4 | 300 | 3 | 4 |

| Metric | Vary FEN and fix IRN | | |
|---|---|---|---|
| | RB + CU | DB + CU | CB + CU |
| PSNR | 37.75 | 37.83 | **37.99** |
| SSIM | 0.960 | 0.960 | **0.962** |

Table 2: Investigation of FEN.

| Metric | Vary IRN and fix FEN | | | |
|---|---|---|---|---|
| | CB + DC | CB + SC | CB + CU$^-$ | CB + CU |
| PSNR | 37.87 | 37.89 | 37.81 | **37.99** |
| SSIM | 0.960 | 0.961 | 0.960 | **0.962** |

Table 3: Investigation of IRN.

the original images to reduce the difference among magnitudes of the four sub-bands. We propose four modes: (1) Original pixel range from 0 to 255. (2) Each pixel divides 255. (3) Each pixel divides 255 and then subtracts the mean of the training dataset by channel. (4) Each pixel divides 255 and then subtracts the mean of the training dataset by channel, then after DWT the coefficients of LL blocks divide a scalar which is around 4 to make the magnitude of LL sub-band more similar to other sub-bands'. The final histograms are showing in the bottom right part of Figure 4. Under the same experiment setting, we pre-process the input images with the four modes. The performance of four modes are shown in the right part of Figure 5. From the figure, we can find that mode 4 gets best performance in terms of loss value. So in the subsequent experiments, we pre-process our input in mode 4.

## 3.2 Investigation of FEN and IRN

To verify the power of the res-clique block and the clique up-sampling module, we designed two contrast experiments. In these two experiments, we used a small version of SRCliqueNet which contains eight blocks, each block having four layers and each layer producing 32 feature maps.In the first experiment, we fixed the clique up-sampling module in IRN and used different blocks, i.e, residual block (RB), dense block (DB) and res-clique block (CB) in FEN. In the second experiment, we fixed the clique blocks in FEN and changed the up-sampling module, i.e, deconvolution (DC), sub-pixel convolution (SC), clique up-sampling without joint learning (CU$^-$) and clique up-sampling (CU). We recorded the best performance in terms of PSNR/SSIM [37] on Set5 with magnification factor $2\times$ during 400 epochs. The performances of all kind of settings are listed in Table 2 and 3.

Table 2 and Table 3 show the power of the clique block and the clique up-sampling module. When we combine them, we get the best performances comparing with other settings.

We also visualize the feature maps of four sub-bands in two stages. Since the channels' number of the two stages is larger than 3, we consider the mean of the feature maps in channel dimension for better visualization, which can be described by $\text{mean}(\mathbf{F}) = \frac{1}{c}\sum_{i=1}^{c}\mathbf{F}_{i,:,:}$. The channel-wise averaged feature maps are shown at the bottom of Figure 3. From Figure 3, we can find that the feature maps of input and stage one do not look like coefficients in the wavelet domain. However, the feature maps of stage two are close to the coefficients of DWT and can reconstruct clear and high resolution images after IDWT. The visualization results demonstrate that it is necessary to add sub-band refinement stage in the clique upsampling module.

## 3.3 Comparison with other wavelet CNN methods

As mentioned above, some exist methods such as Wavelet-SRNet [11] and CNNWSR [21] also used wavelet and CNN for image super-resolution. we first give a detailed comparison with Wavelet-SRNet

and SRCliqueNet. There are three main differences between these two models. (1) Wavelet-SRNet learns wavelet coefficients independently and directly. Our SRCliqueNet considers the relationship among the four sub-bands in the frequency domain. Moreover, Our net applies three stages to learn the coefficients of all sub-bands jointly, i.e. sub-band extraction stage, self residual learning stage and sub-band refinement stage. (2) Wavelet-SRNet uses full wavelet packet decomposition to reconstruct SR images with magnification factor $4\times$ and larger. SRCliqueNet reconstructs SR images with large magnification factor progressively by image pyramid. We use the bicubic down-sampling to resize the ground truth HR image at each level to assist learning. So our net can take full advantage of the supervisory information for HR images. (3) SRCliqueNet is based on clique blocks, which can propagate the information among layers more easily than residual block. We also conduct an experiment to compare these two models on Helen test dataset with magnification factor $4\times$. Our network is trained with images from Helen training dataset, while Wavelet-SRNet is trained with images from both Helen and CelebA datasets. The results are listed in Table 4 below and we can find that our SRCliqueNet outperforms Wavelet-SRNet.

In the following, we list a detailed comparison with CNNWSR and SRCliqueNet. In addition to the above differences between Wavelet-SRNet and SRCliqueNet, CNNWSR is a simpler network with only three layers. CNNWSR supposes that the input LR image is an approximation of LL sub-band. So CNNWSR just tries to learn other three sub-bands by LR image, which is inaccurate. Hence, there is no surprise that our model obviously outstrips CNNSWR in the following quantitative experiment. In [21], the authors show four reconstructed images (names: monarch, zebra, baby and bird) chosen from Set5 and Set14 datasets. The PSNR comparison on these images is shown in Table 5 below.

| Models | PSNR | SSIM | Models | monarch $2\times$ | zebra $2\times$ | baby $4\times$ | bird $4\times$ |
|---|---|---|---|---|---|---|---|
| Wavelet-SRNet [11] | 27.94 | 0.8827 | CNNWSR [21] | 35.74 | 31.84 | 31.58 | 29.01 |
| SRCliqueNet | **28.23** | **0.8844** | SRCliqueNet | **40.53** | **34.71** | **33.90** | **35.84** |

Table 4: Results on Helen test set ($4\times$). Table 5: PSNR comparisons between CNNWSR and SRCliqueNet.

## 3.4 Comparison with the-state-of-the-arts

To validate the effectiveness of the proposed network, we performed several experiments and visualizations. We compared our proposed network with 8 state-of-the-art SR algorithms: DRCN [18], LapSRN [22], DRRN [31], MemNet [32], SRMDNF [42], IDN [14], D-DBPN [8] and EDSR [25]. We carried out extensive experiments using four benchmark datasets mentioned above. We evaluated the reconstructed images with PSNR and SSIM. Table 6 shows quantitative comparisons on $2\times$ and $4\times$ SR. Our SRCliqueNet performs better than existing methods on almost all datasets. In order to maximize the potential performance of our SRCliqueNet, we adopt the self-ensemble strategy similar with [34]. We mark the self-ensemble version of our model as SRCliqueNet+ in Table 6.

In Figure 6, we show visual comparisons on Set14, BSDS100 and Urban100 with a magnification factor $4\times$. Due to limited space, we show only four images results here. For more SR results, please refer to our supplementary materials. As shown in Figure 6, our method accurately reconstructs more clear and textural details of English letters and more textural stripes on zebras. For structured architectural style images, our method tends to get more legible reconstructed HR images. The comparisons suggest that our method infers the high-frequency details directly in the wavelet domain and the results prove its effectiveness. Also, our method gets better quantitative results in terms of PSNR and SSIM than other state-of-the-arts.

## 4 Conclusion

In this paper, we propose a novel CNN called SRCliqueNet for SISR. We design a new up-sampling module called clique up-sampling which uses IDWT to change the size of feature maps and jointly learn all sub-band coefficients depending on the edge feature property. We design a res-clique block to extract features for SR. We verify the necessity of both two modules on benchmark datasets. We also extend our SRCliqueNet with a progressive up-sampling module to deal with larger magnification factors. Extensive evaluations on benchmark datasets demonstrate that the proposed network performs better than the state-of-the-art SR algorithms in terms of quantitative metrics. For visual quality, our algorithm also reconstructs more clear and textual details than other state-of-the-arts.

Table 6: Quantitative evaluation of state-of-the-art SR algorithms: average PSNR/SSIM for magnification factors 2× and 4×. **Red** indicates the best and Blue indicates the second best performance. ('-' indicates that the method failed to reconstruct the whole images due to computation limitation.)

| Models | Mag. | Set5 | | Set14 | | BSDS100 | | Urban100 | |
|---|---|---|---|---|---|---|---|---|---|
| | | PSNR | SSIM | PSNR | SSIM | PSNR | SSIM | PSNR | SSIM |
| Bicubic | 2× | 33.65 | 0.930 | 30.34 | 0.870 | 29.56 | 0.844 | 26.88 | 0.841 |
| VDSR [17] | 2× | 37.53 | 0.958 | 32.97 | 0.913 | 31.90 | 0.896 | 30.77 | 0.914 |
| DRCN [18] | 2× | 37.63 | 0.959 | 32.98 | 0.913 | 31.85 | 0.894 | 30.76 | 0.913 |
| LapSRN [22] | 2× | 37.52 | 0.959 | 33.08 | 0.913 | 31.80 | 0.895 | 30.41 | 0.910 |
| DRRN [31] | 2× | 37.74 | 0.959 | 33.23 | 0.913 | 32.05 | 0.897 | 31.23 | 0.919 |
| MemNet [32] | 2× | 37.78 | 0.960 | 33.28 | 0.914 | 32.08 | 0.898 | 31.31 | 0.920 |
| SRMDNF [42] | 2× | 37.79 | 0.960 | 33.32 | 0.915 | 32.05 | 0.898 | 31.31 | 0.920 |
| IDN [14] | 2× | 37.83 | 0.960 | 33.30 | 0.915 | 32.08 | 0.898 | 31.27 | 0.920 |
| D-DBPN [8] | 2× | 38.09 | 0.960 | 33.85 | 0.919 | 32.27 | 0.900 | - | - |
| EDSR [25] | 2× | 38.11 | 0.960 | 33.92 | 0.919 | 32.32 | 0.901 | **32.93** | 0.935 |
| SRCliqueNet | 2× | **38.23** | **0.963** | **33.96** | **0.923** | **32.36** | **0.905** | 32.86 | **0.936** |
| SRCliqueNet+ | 2× | **38.28** | **0.963** | **34.03** | **0.924** | **32.40** | **0.906** | **32.95** | **0.937** |
| Bicubic | 4× | 28.42 | 0.810 | 26.10 | 0.704 | 25.96 | 0.669 | 23.15 | 0.659 |
| VDSR [17] | 4× | 31.35 | 0.882 | 28.03 | 0.770 | 27.29 | 0.726 | 25.18 | 0.753 |
| DRCN [18] | 4× | 31.53 | 0.884 | 28.04 | 0.770 | 27.24 | 0.724 | 25.14 | 0.752 |
| LapSRN [22] | 4× | 31.54 | 0.885 | 28.19 | 0.772 | 27.32 | 0.728 | 25.21 | 0.756 |
| DRRN [31] | 4× | 31.68 | 0.888 | 28.21 | 0.772 | 27.38 | 0.728 | 25.44 | 0.764 |
| MemNet [32] | 4× | 31.74 | 0.890 | 28.26 | 0.772 | 27.40 | 0.728 | 25.50 | 0.763 |
| SRMDNF [42] | 4× | 31.96 | 0.893 | 28.35 | 0.777 | 27.49 | 0.734 | 25.68 | 0.773 |
| IDN [14] | 4× | 31.82 | 0.890 | 28.25 | 0.773 | 27.41 | 0.730 | 25.41 | 0.763 |
| D-DBPN [8] | 4× | 32.47 | 0.898 | 28.82 | 0.786 | 27.72 | 0.740 | - | - |
| EDSR [25] | 4× | 32.46 | 0.897 | 28.80 | 0.788 | 27.71 | 0.742 | 26.64 | 0.803 |
| SRCliqueNet | 4× | **32.61** | **0.903** | **28.88** | **0.796** | **27.77** | **0.752** | **26.69** | **0.808** |
| SRCliqueNet+ | 4× | **32.67** | **0.903** | **28.95** | **0.797** | **27.81** | **0.752** | **26.80** | **0.810** |

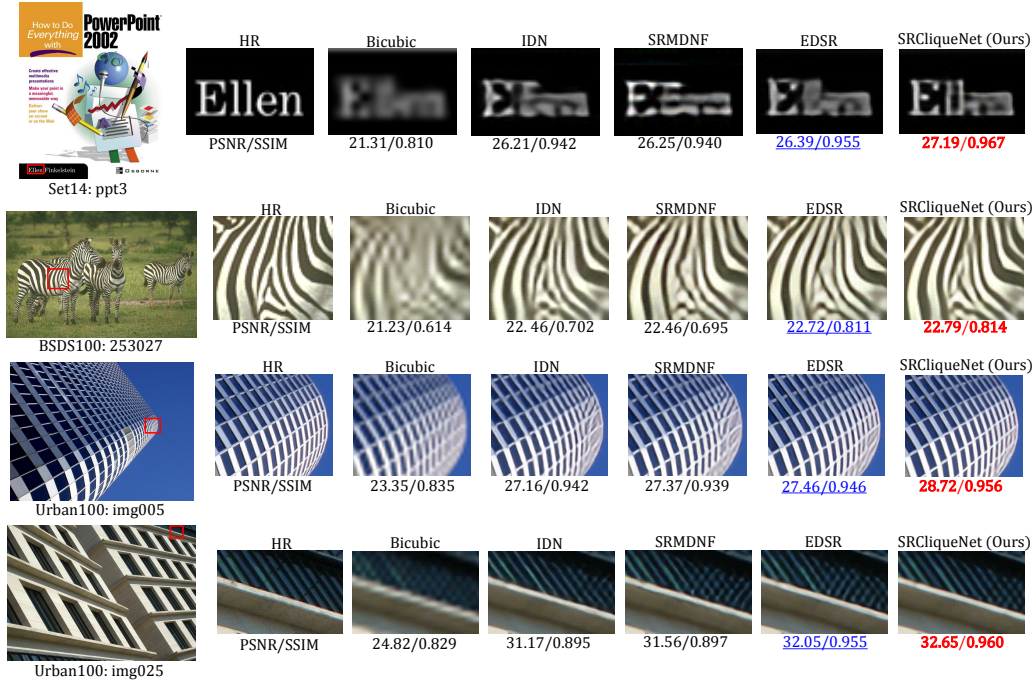

Figure 6: Visual comparisons on images sampled from Set14, BSDS100 and Urban100, with a magnification factor 4×.

## Acknowledgments

This research is partially supported by National Basic Research Program of China (973 Program) (grant nos. 2015CB352502 and 2015CB352303), National Natural Science Foundation (NSF) of China (grant nos. 61625301, 61731018 and 61671027), Qualcomm and Microsoft Research Asia.

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
