[Reviews · NeurIPS 2018]

Reviewer 1



Summary: This paper proposes a CNN architecture called SRCliqueNet for single-image super-resolution (SISR), and it consists of two key parts, feature embedding net (FEN) and the image reconstruction net (IRN). FEN consists of two convolutional layers and a clique block group. The first convolutional layer tries to increase the number of channels of input, which can be added with the output of the clique block group via the skip connection. The second convlutional layer tries to change the number of channels so that they can fit the input of clique block group. The clique block group concatenates features from a sequence of clique blocks, each of which has two stages: 1) the first stage does the same things as dense block, while the second stage distills the feature further. Following the idea of resnet, a clique block contains more skip connections compared with a dense block, so the information among layers can be more easily propagated. IRN consists of a clique up-sampling module and a convolutional layer used to reduce the number of feature maps to reconstruct the HR image with RGB channels. The clique up-sampling module contains four sub networks, representing four sub-bands, LL, LH, HL and HH, in the wavelet domain, respectively. Each sub network has 1) subband extraction, 2) residual block, 3) a refinement stage that uses the high frequency blocks to refine/learn the low frequency blocks, 4) IDWT, and 5) a convolutional layer, which is used to reduce the number of channels and get the HR image. Experiment evaluations on benchmark datasets show that that SRCliqueNet performs better than the state-of-the-art SR algorithms in terms of quantitative metrics. Strengths: - A novel idea of modeling the relationship among the four sub-bands in the clique up-sampling module in IRN. - This paper is technically sound with extensive quantitative evaluation of state-of-the-art SR algorithms on benchmark dataset, and experiment and visualization to demonstrate the usefulness of FEN and IRN. - This paper is well write with clear description of how SRCliqueNet works. Weaknesses: Overall it is a great paper. It will be good if training and testing time is also reported. This is helpful to know if it can be used in real-time SR. Rebuttal Response: After reading the review, I am more clear on the testing time of this approach. For now, it appears that it may not be fast enough to process real time video frame (1 second / image now based on rebuttal), but it does not decrease the value of this work to use DL on wavelet domain. I will remain my score to be 8.

Reviewer 2



This paper presents a method for image super-resolution with a clique network structure. The network has two part, the feature embedded net with clique block (i.e., dense block with skip connection), as well as the clique upsampling net (which borrows the idea from wavelet-based sub-band image processing, by decomposing images into four parts of different frequency (LL, LH, HL, HH)). The final upsampled parts are reconstructed with inverse DWT (haar wavelet) to obtain the high-res image. It is evaluated on some benchmark datasets, and obtained some performance improvement compared to prior work. Overall, the idea of decomposing feature maps into four different frequency part with wavelet is reasonable. The clique block seems the dense block with skip connection, which may be better for preserving high-frequency image details. However, I have a few concerns. 1. The main benefit of deep learning is to learn optimal features/filters from data, rather than relying on hand-crafted features/filters. From this perspective, what is the point to include the inverse DWT (haar wavelet) in the network to reconstruct the image? Isn't it contradicting with itself, since DWT (haar wavelt) is hand-crafted filters/features? As shown in Fig.3 and Fig.4, there should be no surprise that the histograms of the four parts are similar to the those of the wavelet sub-band, because in Fig.3 the IDWT is used to reconstruct the image. 2. The results shown in Table 3 is incremental to me. Compared to the 2nd best result, the improvement of PSNR is less than 0.1db. Even though PSNR may not be a good perceptual-related metric, 0.1db in PSNR is still a very small performance improvement, which may not justify the proposed method. 3. Since [11] and [21] also used wavelet and CNN for image super-resolution, they are closely related to this paper, and should be compared. After reading the rebuttal, I raised my score to 6. If the paper is accepted, please add the comparison results with the two prior work and the clarification of technical details in the final version of the paper.

Reviewer 3



The paper proposes a deep network for image super-resolution using res-clique blocks, already existing in the literature. It is a fairly direct application of the original clique architecture, but seems like a good choice of architecture. The clique blocks are used in two different roles: feature extraction and upsampling. The quantitative and qualitative results show good performance compared with existing baselines.